# Fractal Logistic Equation

## Alireza Khalili Golmankhaneh [1],* and Carlo Cattani [2]

[1] Department of Physics, Urmia Branch, Islamic Azad University, Urmia, PO Box 969, Iran

[2] Engineering School, DEIM, Tuscia University, Viterbo, Italy and Ton Duc Tang University, HCMC, Ho Chí Minh 758307, Vietnam

\* Correspondence: alirezakhalili2002@yahoo.co.in

**Abstract:** In this paper, we give difference equations on fractal sets and their corresponding fractal differential equations. An analogue of the classical Euler method in fractal calculus is defined. This fractal Euler method presets a numerical method for solving fractal differential equations and finding approximate analytical solutions. Fractal differential equations are solved by using the fractal Euler method. Furthermore, fractal logistic equations and functions are given, which are useful in modeling growth of elements in sciences including biology and economics.

**Keywords:** fractal calculus; fractal difference equations; fractal logistic function; fractal logistic equation

## 1. Introduction

Fractal geometry includes shapes which are scale invariant and have fractional dimensions and self-similar properties [1–17]. Analysis on fractals was formulated using different methods such as harmonic analysis, probabilistic methods, measure theory, fractional calculus, fractional spaces, and time-scale calculus [18–29].

Fractional calculus is non-local which provides analogues of local models in science and engineering that found many applications [30–35].

The logistic equation and modified logistic equation have been applied to model growth of population, durable consumer goods, forecasting many social and technological patterns. The growth curves to the biological, technological and economic fields fitted by the logistic function. The differential form of the logistic growth equation was originally developed by Verhulst in 1838 [36]. A modified logistic model was proposed by Blumberg [37]; unlike the original logistic model, this cannot be integrated analytically but only solved numerically. Data analysis fits the modified logistic model [38–40].

Fractional difference and logistic equations were studied by many researchers [41–56].

Recently, in the seminal papers [57,58], generalized standard calculus was adopted to include functions with support on totally disconnected fractal sets and self-avoiding curves. Fractal calculus was applied as a mathematical model for diffraction of light and for random walks which give anomalous diffusion on fractal Cantor-like sets [59–61]. Fractal Sumudu transforms were defined, which are important in control engineering problems [62,63].

In this work, we present that which corresponds to a given fractal difference equation fractal differential equation. We provide analogues of the numerical method for finding the solutions of the fractal differential equations such as the fractal logistic equation.

The framework of the paper is as follows:

In Section 2 we review some basic tools. We define the fractal shift operator and fractal difference operator in Section 3. In Section 4 we use these operators to define fractal difference equations and the corresponding fractal differential equations. The fractal Euler method, which is used as a numerical method for solving fractal differential equations, is given in Section 5. In Section 6 we present the fractal logistic equation and function and conclusions are given in Section 7.

## 2. Basic Tools

In this section, we summarize the local generalized Riemman calculus on the fractal middle-$\kappa$ Cantor set.

### 2.1. Middle-$\kappa$ Cantor Set

The middle-$\kappa$ Cantor set is generated by the following steps:
1-*Step 1.* Pick up an open interval of length $0 < \kappa < 1$ from the middle of the $I = [0, 1]$.

$$C_1^{\kappa} = [0, \frac{1}{2}(1 - \kappa)] \cup [\frac{1}{2}(1 + \kappa), 1]. \tag{1}$$

2-*Step 2.* Delete disjoint open intervals of length $\kappa$ from the middle of the remaining closed intervals of step 1.

$$C_2^{\kappa} = [0, \frac{1}{4}(1 - \kappa)^2] \cup [\frac{1}{4}(1 - \kappa^2), \frac{1}{2}(1 - \kappa)] \cup [\frac{1}{2}(1 + \kappa) + \frac{1}{2}((1 + \kappa)$$
$$+ \frac{1}{2}(1 - \kappa)^2)] \cup [\frac{1}{2}(1 + \kappa)(1 + \frac{1}{2}(1 - \kappa)), 1]. \tag{2}$$

3-*Step m.* Remove disjoint open intervals of length $\kappa$ from the middle of the remaining closed intervals of step $m$-1.

$$C^{\kappa} = \bigcap_{m=1}^{\infty} C_m^{\kappa}. \tag{3}$$

Note that the measure of middle-$\kappa$ Cantor sets is zero [64].

For every middle-$\kappa$ Cantor set which generally is called Cantor-like sets, the Hausdorff dimension is given by

$$\dim_H(C^{\kappa}) = \frac{\log 2}{\log 2 - \log(1 - \kappa)}, \tag{4}$$

where $H(C^{\kappa})$ is the Hausdorff measure which was used to derive Hausdorff dimension [64].

### 2.2. Local Fractal Calculus

If $C^{\kappa}$ is middle-$\kappa$ Cantor set then the flag function is defined by [57,58,60],

$$F(C^{\kappa}, J) = \begin{cases} 1, & \text{if } C^{\kappa} \cap J \neq \emptyset; \\ 0, & \text{otherwise,} \end{cases} \tag{5}$$

where $J = [b_1, b_2]$. Then, $\mathcal{P}^{\alpha}[C^{\kappa}, W]$ is given in [57,58,60] by

$$\mathcal{P}^{\alpha}[C^{\kappa}, W] = \sum_{i=1}^{n} \Gamma(\alpha + 1)(t_i - t_{i-1})^{\alpha} F(C^{\kappa}, [t_{i-1}, t_i]), \tag{6}$$

where $W_{[b_1, b_2]} = \{b_1 = t_0, t_1, t_2, \ldots, t_n = b_2\}$ is a subdivisions of $J$.

The **mass function** $\gamma^{\alpha}(C^{\kappa}, b_1, b_2)$ is defined in [57,58,60] by

$$\gamma^{\alpha}(C^{\kappa}, b_1, b_2) = \lim_{\delta \to 0} \gamma_{\delta}^{\alpha} = \lim_{\delta \to 0} \left( \inf_{E_{[r,t]}: |E| \leq \delta} \mathcal{P}^{\alpha}[C^{\kappa}, E] \right), \tag{7}$$

here, infimum is taking over all subdivisions $E$ of $[b_1, b_2]$ satisfying $|E| := \max_{1 \leq i \leq n}(t_i - t_{i-1}) \leq \delta$.

The **integral staircase function** $S_{C^\kappa}^\alpha(t)$ is defined in [57,58] by

$$
S_{C^\kappa}^\alpha(t) = \begin{cases} \gamma^\alpha(C^\kappa, b_0, t), & \text{if} \quad t \geq b_0; \\ -\gamma^\alpha(C^\kappa, b_0, t), & \text{otherwise}, \end{cases}
\tag{8}
$$

where $b_0$ is an arbitrary and fixed real number.

The **$\gamma$-dimension** of a set $C^\kappa \cap [b_1, b_2]$ is defined

$$
\begin{aligned}
\dim_\gamma(C^\kappa \cap [b_1, b_2]) &= \inf\{\alpha : \gamma^\alpha(C^\kappa, b_1, b_2) = 0\} \\
&= \sup\{\alpha : \gamma^\alpha(C^\kappa, b_1, b_2) = \infty\}.
\end{aligned}
\tag{9}
$$

The **$C^\alpha$-limit** of a function $g : \Re \to \Re$ is given by

$$
\forall\, \epsilon > 0,\ \exists\, \delta > 0\ z \in C^\kappa \quad \text{and} \quad |z - t| < \delta \Rightarrow |g(z) - \ell| < \epsilon.
\tag{10}
$$

If $\ell$ exists, then we have

$$
\ell = C^\alpha - \lim_{z \to t} g(z).
\tag{11}
$$

The **$C^\alpha$-continuity** of a function $k : \Re \to \Re$ is defined by

$$
g(t) = C^\alpha - \lim_{z \to t} g(z).
\tag{12}
$$

The **$C^\alpha$-derivative** of $f(t)$ at $t$ is defined [57]

$$
D_{C^\kappa}^\alpha f(t) = \begin{cases} C^\alpha - \lim_{y \to t} \dfrac{f(y) - f(t)}{S_{C^\kappa}^\alpha(y) - S_{C^\kappa}^\alpha(t)}, & \text{if} \quad t \in C^\kappa; \\ 0, & \text{otherwise}. \end{cases}
\tag{13}
$$

if the limit exists.

The **$C^\alpha$-integral** of $k(t)$ on $J = [b_1, b_2]$ is defined in [57,58,60] and approximately given by

$$
\int_{b_1}^{b_2} f(t) d^\alpha t \approx \sum_{i=1}^{n} f(t_i)(S_{C^\kappa}^\alpha(t_i) - S_{C^\kappa}^\alpha(t_{i-1})).
\tag{14}
$$

For more details we refer the reader to [57,58].

The **Characteristic function of the middle-$\kappa$ Cantor set** is defined in [60] by

$$
\chi_{C^\kappa}(\alpha, t) = \begin{cases} \dfrac{1}{\Gamma(\alpha+1)}, & t \in C^\kappa; \\ 0, & \text{otherwise}. \end{cases}
\tag{15}
$$

**Some important formulas:**

$$
\begin{aligned}
D_{C^\kappa}^\alpha \chi_{C^\kappa}(\alpha, t) t &= \frac{1}{\Gamma(\alpha+1)} \chi_{C^\kappa}(\alpha, t), \\
D_{C^\kappa}^\alpha \chi_{C^\kappa}(\alpha, t) t^2 &= \frac{2}{\Gamma(\alpha+1)} \chi_{C^\kappa}(\alpha, t) t, \\
D_{C^\kappa}^\alpha \sin(t \chi_{C^\kappa}(\alpha, t)) &= \frac{1}{\Gamma(\alpha+1)} \cos(t \chi_{C^\kappa}(\alpha, t)).
\end{aligned}
$$

In Figure 1 we have clarified the definitions of Section 2.

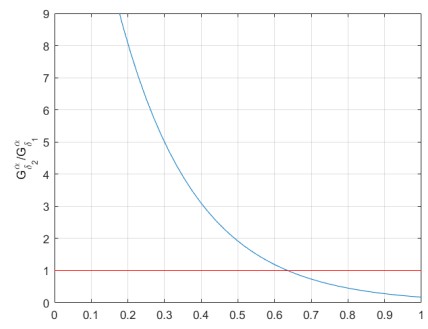

(**a**) Middle-$\kappa$ Cantor set with $\kappa = 1/3$

(**b**) Staircase function corresponding to middle-$\kappa$ Cantor set with $\kappa = 1/3$



(**c**) The $\gamma$-dimension gives $\alpha = 0.5$ to middle-$\kappa$ Cantor set with $\kappa = 1/3$

(**d**) Characteristic function for middle-$\kappa$ Cantor set with $\kappa = 1/3$

**Figure 1.** Figures for the Section 2.

## 3. Fractal Finite Difference and Fractal Derivative

In this section, we define fractal shift operator and fractal difference operator. Furthermore, the fractal differential equation and its corresponding difference equation is given. Let us define a $\alpha$-difference equation on the fractal middle-$\eta$ Cantor set as follows:

$$z_{K,n+1} = \chi_K B\, z_{K,n},$$

where $C^\kappa = K$ and $B$ is an operator. In view of fractal Taylor expansion, we have

$$
\begin{aligned}
z_{K,n+1} &= \sum_{k=0}^{\infty} \frac{S_K^\alpha(h)^k}{k!} (D_K^\alpha)^k z(t)|_{t=t_n} \\
&= e^{S_K^\alpha(h) D_K^\alpha} z_n,
\end{aligned}
\tag{16}
$$

where

$$
\begin{aligned}
z_{n+1} &= z(t_{n+1}), \\
t_{n+1} &= h + t_n, \quad h \in \Re, \quad z_n \in K, \\
D_K^\alpha z_{K,n} &= D_K^\alpha z_K(t)|_{t=t_n},
\end{aligned}
\tag{17}
$$

consequently, we have

$$B = e^{S_K^\alpha(h) D_K^\alpha}.
\tag{18}$$

In view of Equation (16), $B$ is called **fractal shift operator**.

**Fractal difference operator** is denoted by $\Delta_K$ and defined by

$$\Delta_K = B - 1, \tag{19}$$

it follows that

$$\Delta_K z_n = z_{n+1} - z_n. \tag{20}$$

Using Equation (18) the fractal $\alpha$-order derivative in terms of the fractal $\alpha$-difference equation is

$$
\begin{aligned}
D_K^\alpha &= \frac{1}{S_K^\alpha(h_n)} \ln B \\
&= \frac{1}{S_K^\alpha(h)} \ln(1 + \Delta_K) \\
&= \frac{1}{S_K^\alpha(h)} \sum_{k=1}^\infty (-1)^{k+1} \frac{\Delta_K^k}{k!}.
\end{aligned}
$$

Fractal higher $\alpha$-order derivatives is approximated by

$$(D_K^\alpha)^m \approx \left(\frac{\Delta_K}{S_K^\alpha(h)}\right)^m. \tag{21}$$

**Example 1.** *Consider $z(t) = S_K^\alpha(t)$, then using Equation (16) we can write*

$$
\begin{aligned}
S_{K,n+1}^\alpha &= \chi_K B S_{K,n}^\alpha \\
&= \sum_{k=0}^\infty \frac{S_K^\alpha(h)^k}{k!} (D_K^\alpha)^k S_K^\alpha(t)|_{t=t_n} \\
&= S_K^\alpha(t_n) + S_K^\alpha(h)\chi_K(t_n),
\end{aligned}
$$

*where*

$$D_K^\alpha S_{K,n}^\alpha = D_K^\alpha S_K^\alpha(t)|_{t=t_n} = \chi_K(t_n). \tag{22}$$

*By Equation (20) we have*

$$\Delta_K S_{K,n}^\alpha = S_{K,n+1}^\alpha - S_{K,n}^\alpha. \tag{23}$$

*Using Equation (21) we get*

$$
\begin{aligned}
(D_K^\alpha)^2 S_K^\alpha(t) &= \frac{1}{S_K^\alpha(h)^2} (\Delta_K)^2 S_{K,n}^\alpha \\
&= \frac{1}{S_K^\alpha(h)^2} \Delta_K (S_{K,n+1}^\alpha - S_{K,n}^\alpha) \\
&= \frac{1}{S_K^\alpha(h)^2} (S_{K,n+2}^\alpha - 2S_{K,n+1}^\alpha + S_{K,n}^\alpha). 
\end{aligned}
\tag{24}
$$

*We note that Equation (24) is used to obtain numerical solution of the second $\alpha$-order fractal differential equations.*

## 4. Fractal Difference and Differential Equations

In this section, we show the relation between fractal difference equations and fractal differential equations which should later be used to find numerical solution of the fractal differential equations.

Consider the following fractal $\alpha$-difference equation

$$z_{n+1} = \chi_K \mathbf{B} z_n, \quad z_n \in K, \quad \mathbf{B} \in \Re, \tag{25}$$

then the solution of Equation (25) is

$$z_n = \chi_K \mathbf{B}^n z_0. \tag{26}$$

If we set $z_n = z(t_n)$, then

$$S_K^\alpha(t_{n+1}) = S_K^\alpha(t_n) + S_K^\alpha(h), \quad t_n, \quad t_{n+1} \in K, \quad h \in (0,1), \tag{27}$$

and

$$S_K^\alpha(t_n) = n S_K^\alpha(h). \tag{28}$$

It follows that Equation (25) gives

$$\frac{z(t_n + h) - z(t_n)}{S_K^\alpha(h)} = \frac{\mathbf{B} - 1}{S_K^\alpha(h)} z(t_n), \tag{29}$$

then the corresponding fractal differential equation is

$$D_K^\alpha z(t) = \frac{\mathbf{B} - 1}{S_K^\alpha(h)} z(t). \tag{30}$$

The solution of Equation (30) is

$$z(t) = z(0) \exp\left(\frac{\mathbf{B} - 1}{S_K^\alpha(h) S_K^\alpha(t)}\right). \tag{31}$$

## 5. Numerical Method for Solving Fractal Differential Equation

In this section, we present an analogue of Euler method in the fractal calculus by using fractal difference operator.

Let us consider $\alpha$-order fractal differential equation as

$$D_K^\alpha z(t) = f(S_K^\alpha(t), z(t)), \quad z(t_0) = z_0. \tag{32}$$

In view of Equation (21) we obtain

$$z_{n+1}(t) = z_n(t) + S_K^\alpha(h) f(S_K^\alpha(t), z(t)). \tag{33}$$

By substituting fractal Taylor expansion of $f(S_K^\alpha(t), z(t))$, in Equation (33) we get approximate analytical solution of Equation (32) as follows:

$$z(t + h) = z(t_0) + S_K^\alpha(h) D_K^\alpha z(t) + \frac{1}{2} S_K^\alpha(h)^2 (D_K^\alpha)^2 z(t) + O(S_K^\alpha(h)^3). \tag{34}$$

The numerical solution of the fractal differential equation is obtained by utilizing Equation (33), which might be called **fractal Euler method**.

Considering conjugacy of ordinary calculus and fractal calculus, the **fractal local truncation error** (FLTE) is given by

$$FLTE = z(t_0 + h) - z(t_1) = \frac{1}{2} S_K^\alpha(h)^2 (D_K^\alpha)^2 z(t)|_{t=t_0} + O(S_K^\alpha(h)^3), \tag{35}$$

Equation (35) is valid, if we have

$$\forall t \in K, \quad \exists M > 0, \quad |(D_K^\alpha)^3 z(t)| < M. \tag{36}$$

Recalling conjugacy of ordinary calculus and fractal calculus, the **fractal global truncation error** is given by

$$\forall\, t \in K,\ \exists\, N > 0,\ |(D_K^\alpha)^2 z(t)| < N, \tag{37}$$

and $f$ is the **fractal Lipschitz continuous**, namely

$$|f(S_K^\alpha(t), z_1) - f(S_K^\alpha(t), z_2)| < Q|z_1 - z_2|. \tag{38}$$

In the same manner, the fractal bounded global truncation error (FGTE) is given by

$$|FGTE| \leq \frac{S_K^\alpha(h)N}{2Q}\left(\exp(Q(S_K^\alpha(t) - S_K^\alpha(t_0)) - 1)\right), \tag{39}$$

since $S_K^\alpha(h) \leq h^\alpha$ then we obtain

$$|FGTE| \approx h^\alpha. \tag{40}$$

For this reason, the fractal Euler method is also called $\alpha$-order Euler method.

**Example 2.** *Consider the following fractal Cauchy problem*

$$D_K^\alpha z(t) = z(t), \quad z(0) = 1. \tag{41}$$

*The exact solution Equation* (41) *is*

$$z(t) = \exp(S_K^\alpha(t)). \tag{42}$$

*Applying Equation* (34), *we proceed with corresponding difference equation as follows*

$$z_{n+1}(t) = z_n(t) + S_K^\alpha(h)z(t), \tag{43}$$

*so that by using the fractal Euler method to approximate solution of Equation* (41) *we get*

$$z(t + h) = 1 + S_K^\alpha(h)z(t) + \frac{1}{2}S_K^\alpha(h)^2(D_K^\alpha)^2 z(t) + O(S_K^\alpha(h)^3). \tag{44}$$

In Figure 2 as well as in the following figures, the smooth colored lines represent the standard results with $\alpha = 1$.

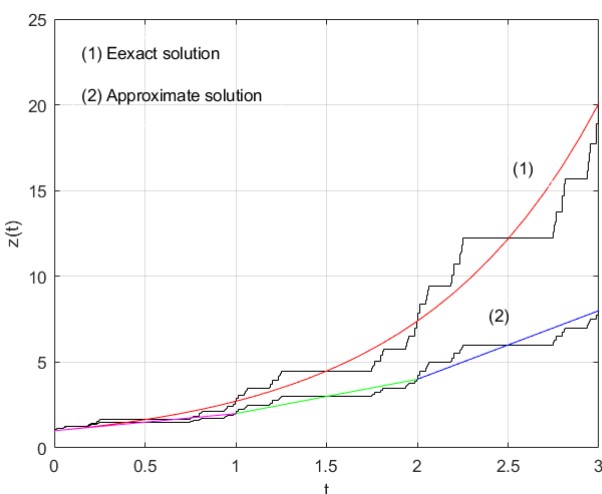

**Figure 2.** Exact solution and approximate solution using numerical fractal Euler method where step sizes $h = 1$ and $\alpha = 0.63$.

**Example 3.** *Consider the fractal time varying harmonic oscillator as follows*

$$(D_K^\alpha)^2 z(t) + z(t) = 0, \quad z(0) = 1, \quad D_K^\alpha z(t)|_{t=0}. \tag{45}$$

*In view of conjugacy of fractal and standard calculus, the solution of Equation* (45) *is* $z(t) = \cos(S_K^\alpha(t))$. *The corresponding fractal difference equation is*

$$\frac{1}{S_K^\alpha(h)^2}(z_{n+2} - 2z_{n+1} + z_n) + z_n = 0, \tag{46}$$

*with initial conditions*

$$z_0 = 0, \, z_1 = 0. \tag{47}$$

Using the standard techniques one can get the solution of recurrence relations Equation (46) as follows

$$z_n = \frac{1}{2}((1 + iS_K^\alpha(h))^n + (1 - iS_K^\alpha(h))^n), \tag{48}$$

where $z_n$ is real number and $i = \sqrt{-1}$. By Equation (28) we have

$$z_n = \cos(S_K^\alpha(t_n)) + \frac{1}{2}S_K^\alpha(h)S_K^\alpha(t_n)\cos(S_K^\alpha(t_n)) + O(S_K^\alpha(h)^2). \tag{49}$$

The approximate solution of Equation (46) for the case $S_K^\alpha(t_n)S_K^\alpha(h) \ll 1$ is

$$z_n = \cos(S_K^\alpha(t_n)), \tag{50}$$

and

$$S_K^\alpha(h) \to 0 \Rightarrow z_n = \cos(S_K^\alpha(t_n)). \tag{51}$$

It is expected.

## 6. Fractal Logistic Equation

In this section we give the fractal logistic equation and its solution.

Let us consider fractal logistic equation as follows:

$$D_{K,t}^\alpha z(t) = r_K z(t)\left(1 - \frac{z(t)}{r_K'}\right), \tag{52}$$

where $r_K$ is called **fractal growth parameter** and $r_K'$ is called **fractal carrying capacity**. Applying conjugacy of fractal calculus and standard calculus we obtain the solution of Equation (52) as follows:

$$z(t) = \frac{z(0)r_K}{z(0) + (r_K - z(0))\exp(-r_K' S_K^\alpha(t))}, \tag{53}$$

where $z(0)$ is called initial population. If we consider upper limit of $S_K^\alpha(t) < t^\alpha$ we get

$$z(t) \simeq \frac{z(0)r_K}{z(0) + (r_K - z(0))\exp(-r_K' t^\alpha)}, \tag{54}$$

where $z(t)$ is called **fractal Logistic function**. In Figure 3 we have plotted Equation (53).

The fractal derivative of Equation (53) is

$$D_{K,t}^\alpha z(t) = \frac{z(0)r_K(r_K - z(0))r_K'\exp(-r_K' S_K^\alpha(t))}{(z(0) + (r_K - z(0))\exp(-r_K' S_K^\alpha(t)))^2}, \tag{55}$$

and its upper bound is

$$D^\alpha_{K,t}z(t) \simeq \frac{z(0)r_K(r_K - z(0))r'_K \exp(-r'_K t^\alpha)}{(z(0) + (r_K - z(0))\exp(-r'_K t^\alpha))^2}. \tag{56}$$

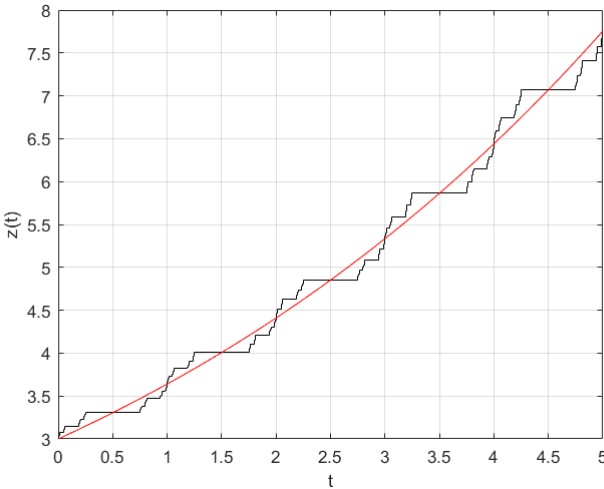

**Figure 3.** Fractal logistic curve with $\alpha = 0.5$ where $\mu = 1/2$. $z(0) = 3$, $r_K = 100$, $r'_K = 0.2$.

In Figure 4, we present the upper bound of fractal logistic function and its fractal derivatives by setting different value of $\alpha$.

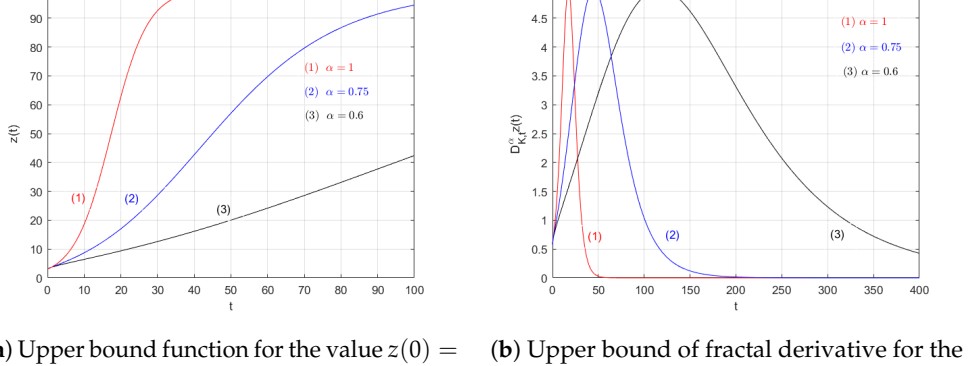

(**a**) Upper bound function for the value $z(0) = 3$, $r_K = 100$, $r'_K = 0.2$

(**b**) Upper bound of fractal derivative for the values $z(0) = 3$, $r_K = 100$, $r'_K = 0.2$

**Figure 4.** Graph of upper bound of fractal logistic function and its fractal derivative.

The inflection point $(t = t_{ip})$ of fractal Logistic function is defined $z(t_{ip}) = r_K/2$, then we have

$$S^\alpha_K(t_{ip}) = \frac{\ln(r_K/z(0) - 1)}{r'_K}. \tag{57}$$

Using the upper bound of $S^\alpha_K(t) \le t^\alpha$, we get

$$t_{ip}(\alpha) = \left( \frac{\ln(r_K/z(0) - 1)}{r'_K} \right)^\alpha. \tag{58}$$

In Figure 5 we have sketched Equation (58) which indicates inflection points versus $\alpha$.

**Remark:** *In figures of this section, we have shown that the power law model for the processes with the fractal structure by using fractal calculus leads to ordinary cases by letting $\alpha = 1$.*

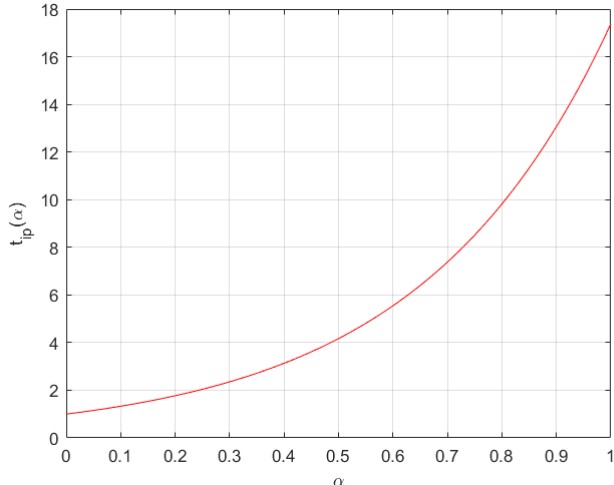

**Figure 5.** The inflection point as function of $\alpha$ by choosing $z(0) = 3$, $r_K = 100$, $r'_K = 0.2$.

## 7. Conclusions

In this paper, we have defined fractal difference equations which are useful for finding numerical methods for solving fractal differential equations. The fractal differential equations and corresponding difference equations were given. Fractal logistic equations and functions were suggested. The upper bound of the fractal logistic function can be considered the most generalized growth model by changing different fractal dimensions. A function of generalized inflection points in terms of fractal dimension was obtained. We have also solved illustrative examples.

**Author Contributions:** All authors worked jointly and contributed equally.

**Funding:** This research received no external funding.

**Conflicts of Interest:** The authors declare no conflict of interest.

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
