# Peer review of "Fractal Logistic Equation"

_fractalfract, doi:10.3390/fractalfract3030041_

Round 1
Reviewer 1 Report
The paper is very good contribution, but spelling mistakes must be removed
Author Response
We are grateful to referee for evaluation of our work.
We hope that the revised manuscript meets the expectations of referee.
Sincerely,
Prof. Alireza K. Golmankhaneh
Prof Carlo Cattani
Reviewer 2 Report
This is a useful contribution. Here authors studied difference equations on fractal sets and their corresponding fractal differential equations. An analogue of the classical Euler method in fractal calculus is defined. This fractal Euler method presets a numerical method for solving fractal differential equations and finding approximate analytical solutions. Fractal differential equations are solved by using the fractal Euler method. Furthermore, fractal logistic equations and functions are given, which are useful in modeling growth of elements in sciences including biology and economics. The results are new and interesting. Thus, I recommend this paper for publication after some minor revisions. 1. Overall presentation of the paper needs to improve. 2. In numerical section a few more comments about the differences between all the Figs. would be more interesting for the readers. 3. Please check the manuscript carefully for typos and grammar errors. 4. Improving the introduction and discussing about the new development in fractional calculus in detail. The following reference may be useful: * A new analysis for fractional rumor spreading dynamical model in a social network with Mittag-Leffler law, Chaos 29 (2019) 013137. *New aspects of fractional Biswas-Milovic model with Mittag-Leffler law, Mathematical Modelling of Natural Phenomena 14(3) 303 (2019). * On the local fractional wave equation in fractal strings, Mathematical Methods in the Applied Sciences 42(5) (2019), 1588-1595 * An efficient numerical algorithm for the fractional Drinfeld-Sokolov-Wilson equation, Applied Mathematics and Computation 335 (2018), 12-24. * A fractional epidemiological model for computer viruses pertaining to a new fractional derivative, Applied Mathematics and Computation 316 (2018) 504-515. After above minor revisions, I strongly recommended this paper for publication in your journal.
Author Response
We are grateful to referee for the thoughtful and careful evaluation of our work.
This is a useful contribution. Here authors studied difference equations on fractal sets and their corresponding fractal differential equations. An analogue of the classical Euler method in fractal calculus is defined. This fractal Euler method presets a numerical method for solving fractal differential equations and finding approximate analytical solutions. Fractal differential equations are solved by using the fractal Euler method. Furthermore, fractal logistic equations and functions are given, which are useful in modeling growth of elements in sciences including biology and economics. The results are new and interesting. Thus, I recommend this paper for publication after some minor revisions. 1. Overall presentation of the paper needs to improve. 2. In numerical section a few more comments about the differences between all the Figs. would be more interesting for the readers. 3. Please check the manuscript carefully for typos and grammar errors. 4. Improving the introduction and discussing about the new development in fractional calculus in detail. The following reference may be useful: * A new analysis for fractional rumor spreading dynamical model in a social network with Mittag-Leffler law, Chaos 29 (2019) 013137. *New aspects of fractional Biswas-Milovic model with Mittag-Leffler law, Mathematical Modelling of Natural Phenomena 14(3) 303 (2019). * On the local fractional wave equation in fractal strings, Mathematical Methods in the Applied Sciences 42(5) (2019), 1588-1595 * An efficient numerical algorithm for the fractional Drinfeld-Sokolov-Wilson equation, Applied Mathematics and Computation 335 (2018), 12-24. * A fractional epidemiological model for computer viruses pertaining to a new fractional derivative, Applied Mathematics and Computation 316 (2018) 504-515. After above minor revisions, I strongly recommended this paper for publication in your journal.
Answer: In the revised manuscript we have considered all comments.
We hope that the revised manuscript meets the expectations of referee and is acceptable for publication.
Sincerely,
Prof. Alireza K. Golmankhaneh
Prof Carlo Cattani
Reviewer 3 Report
see the attached file.

Author Response
We are grateful to referee for the thoughtful and careful evaluation of our work.
Answer: We have revised the manuscript by considereing all comments.
We hope that the revised manuscript meets the expectations of referee and is acceptable for publication.
Sincerely,
Prof. Alireza K. Golmankhaneh
Prof Carlo Cattani

Round 2
Reviewer 3 Report
Accept.